# External Validation of Prediction Models for Surgical Complications in People Considering Total Hip or Knee Arthroplasty Was Successful for Delirium but Not for Surgical Site Infection, Postoperative Bleeding, and Nerve Damage: A Retrospective Cohort Study

**DOI:** 10.3390/jpm13020277

**Published:** 2023-01-31

**Authors:** Lieke Sweerts, Pepijn W. Dekkers, Philip J. van der Wees, Job L. C. van Susante, Lex D. de Jong, Thomas J. Hoogeboom, Sebastiaan A. W. van de Groes

**Affiliations:** 1Department of Orthopaedics, Radboud Institute for Health Sciences, Radboud University Medical Center, 6500 HB Nijmegen, The Netherlands; 2IQ Healthcare, Radboud Institute for Health Sciences, Radboud University Medical Center, 6500 HB Nijmegen, The Netherlands; 3Department of Rehabilitation, Radboud Institute for Health Sciences, Radboud University Medical Center, 6500 HB Nijmegen, The Netherlands; 4Department of Orthopedics, Rijnstate Hospital, 6800 TA Arnhem, The Netherlands

**Keywords:** decision support techniques, external validation, prediction, surgical complications, total hip arthroplasty, total knee arthroplasty

## Abstract

Although several models for the prediction of surgical complications after primary total hip or total knee replacement (THA and TKA, respectively) are available, only a few models have been externally validated. The aim of this study was to externally validate four previously developed models for the prediction of surgical complications in people considering primary THA or TKA. We included 2614 patients who underwent primary THA or TKA in secondary care between 2017 and 2020. Individual predicted probabilities of the risk for surgical complication per outcome (i.e., surgical site infection, postoperative bleeding, delirium, and nerve damage) were calculated for each model. The discriminative performance of patients with and without the outcome was assessed with the area under the receiver operating characteristic curve (AUC), and predictive performance was assessed with calibration plots. The predicted risk for all models varied between <0.01 and 33.5%. Good discriminative performance was found for the model for delirium with an AUC of 84% (95% CI of 0.82–0.87). For all other outcomes, poor discriminative performance was found; 55% (95% CI of 0.52–0.58) for the model for surgical site infection, 61% (95% CI of 0.59–0.64) for the model for postoperative bleeding, and 57% (95% CI of 0.53–0.61) for the model for nerve damage. Calibration of the model for delirium was moderate, resulting in an underestimation of the actual probability between 2 and 6%, and exceeding 8%. Calibration of all other models was poor. Our external validation of four internally validated prediction models for surgical complications after THA and TKA demonstrated a lack of predictive accuracy when applied in another Dutch hospital population, with the exception of the model for delirium. This model included age, the presence of a heart disease, and the presence of a disease of the central nervous system as predictor variables. We recommend that clinicians use this simple and straightforward delirium model during preoperative counselling, shared decision-making, and early delirium precautionary interventions.

## 1. Introduction

Discussing the risk of surgical complications with patients is an important part of shared decision-making in patients with end-stage hip or knee osteoarthritis considering total hip or knee arthroplasty (THA or TKA). Previous research demonstrates that surgical complications are associated with factors relating to demography, comorbidities, and medication use [1], and that these factors combined might predict the risk of surgical complications [2,3,4,5]. Prediction models provide an estimate of a patient’s preoperative risk by calculating individual predicted probabilities for surgical complications, and can thereby be used to facilitate preoperative personalized counselling and shared decision-making.

Although several prediction models for preoperative counselling regarding the risk of surgical complications are available [2,4,5,6,7,8,9,10,11,12,13,14,15], only two studies validated these models externally for patients considering primary THA and TKA [3,16]. External validation is important because differences in, for example, population characteristics and setting may affect the applicability of the prediction models in another hospital population [17]. The available studies that did evaluate the external validation of preoperative prediction models regarding the risk of surgical complications after THA and TKA found moderate-to-poor predictive performance in a new context and patient population, rendering these models unfit for application in clinical practice [3,16]. These results emphasize the need for further research in the development of accurate risk stratification tools specified for people opting for THA or TKA.

Only three studies with three procedure-specific prediction models to predict the risk on a surgical site infection after primary THA or TKA have been published [2,14,15], and to our knowledge, none of these models have been externally validated [18]. Focusing upon one of these studies, our own research group recently published a set of easy to use and potentially valid prediction models [2]. These prediction models have been developed specifically for the preoperative counselling of four surgical complications: surgical site infection, postoperative bleeding, delirium, and nerve damage after primary THA and TKA. These four models are considered to have good applicability since the models comprised less than eight predictor variables, which is considered easy to use in clinical practice [19]. Furthermore, the models in this study showed moderate-to-good discriminative capacity, and are considered to be valuable in predicting these surgical complaints since current counselling is based on population-based risks [19]. However, these models were only validated internally on the basis of data from a single academic hospital setting in the Netherlands [2], which generally is a setting which includes a more complex patient population. Furthermore, the applicability of the models has not yet been confirmed using data of patients from another setting, and the models’ predictive performance using a non-academic hospital population has not been tested. In other words, the performance of these prediction models in an external population is unknown; this is important to determine whether the models are transferrable to a broader context. The aim of this study was to determine the external validity of four previously developed models for surgical complications in people considering primary THA or TKA surgery by determining the predictive performance in this new context.

## 2. Materials and Methods

### 2.1. Study Design, Setting, and Population

In this retrospective cohort study, four prediction models developed previously by Sweerts et al. were externally validated using data from a cohort of patients from Rijnstate Hospital in Arnhem, the Netherlands [2]. We considered this cohort as being representative of a non-academic hospital population in the Netherlands. The prediction models were originally developed for patients considering primary THA or TKA. As such, patients for our current study were eligible for inclusion if they had had a THA or TKA for the first time. Patients who had revision arthroplasty of one or several components of the joint prosthesis were excluded because research has shown that revision surgery increases the risk for surgical complications [2,20,21].

All patients that underwent primary THA or TKA between 2017 and 2020 and met the inclusion criteria were contacted to ask for their consent to use their pseudonymized patient data.

Approval for this study was granted by the Institutional Research Board of Rijnstate Hospital (2020-1584). The study was performed and reported in line with transparent reporting of a multivariable prediction model for individual prognosis or diagnosis (TRIPOD-) guidelines [22].

### 2.2. Data Extraction and Handling

Data of variables used in the prediction models were extracted from the patients’ electronic health records using CTcue, a self-service data mining tool with text-mining features (CTcue, v4.4.1; Amsterdam, The Netherlands, www.ctcue.com, accessed on 12 July 2022). This clinical data collection tool is powered by artificial intelligence and machine learning to parse, structure, and interpret data. The tool adheres to the General Data Protection Regulation and the program uses pseudo-identification to ensure patient privacy [23]. Identifiers that could be linked to individual patients (e.g., names, addresses, phone numbers) were pseudonymized. The tool was used to extract both structured (e.g., age, measurement values, standardized diagnostic codes, test results) and unstructured (free texts such as the physician’s medical notes and evaluations) patient data from electronic health record systems by use of a query. The query used in this study was based on specification of the category to extract data from the electronic health record, and was further specified by filtering on specific report type, period, specialism, date, etc. A variety of categories of the electronic health record system can be searched as reports, medication administrations, appointments, care activities, surgeries, vital signs, etc. [23]. The program collates these data in an analyzable dataset [23]. Data were extracted by combining keywords with commonly known synonyms, variants, abbreviations, and frequent typographical errors as suggested by the application programming interface. One researcher (LS) created the query and checked all the extracted data using the validation tool of the clinical data collection tool. The query for this study was initially performed with high sensitivity. The query was later narrowed while checking whether this would not lead to any data loss. The query was fine-tuned to the point where no new information was found. The query used can be found in Appendix A. Structured data were considered missing if the outcome variable was not available from the patients’ electronic health record. These missing data were first checked for patterns of randomness, and subsequently imputed by multiple imputation, using predictive mean matching. The number of imputations was set to ten. The imputation was checked for accuracy by visual inspection and frequencies. Specific outcome variables not reported in the unstructured data (e.g., no comments about infections or comorbid conditions in the free text fields) were considered as a sign that these were also not present in that particular patient. After the search using CTcue, another researcher (PD) extracted all relevant data from the electronic health records manually. Both researchers subsequently randomly checked the accuracy of 100 patient records by comparing the clinical data collection tool and manual data extraction. The agreement between the automated and manual search was measured by Cohen’s κ coefficient, with κ = 0.41–0.60 indicating moderate agreement, κ = 0.61–0.80 representing good agreement, and κ ≥ 0.81 representing very good agreement [24].

### 2.3. Predictor Variables

In line with the previously developed models, we extracted the following variables: age, gender, BMI, smoking status (yes/no), the presence of predefined comorbidities (yes/no), and predefined medication use (yes/no) [2]. Comorbidities included the presence of an immunological disorder, rheumatoid arthritis, diabetes mellitus, liver disease, heart disease, disease of the central nervous system, and/or hip dysplasia. Information collected regarding medication use included the of use of vitamin K antagonists, and/or non-steroid anti-inflammatory drugs (NSAID) [2].

### 2.4. Predictor Variables

The outcome variables used for the external validation of the prediction models included the presence (yes/no) of surgical site infection within 90 days after surgery, postoperative bleeding, delirium and nerve damage [2]. Only models with a mean AUC >0.7 in the developmental phase were considered appropriate for external validation. Therefore, the models for venous thromboembolism and luxation were not included in this study.

### 2.5. Sample Size

The sample size was based on the rule of thumb that at least five events per variable are required for each predictor in the models [25]. An event was defined as the postoperative occurrence of one of the predefined surgical complications. In the Netherlands, the risk of a surgical complication such as surgical site infection is 3% [26]. As the prediction model for surgical site infection consists of six variables, a sample size of at least 1000 patients was required (6 × 5/0.03 = 1000).

### 2.6. Data Analysis

#### 2.6.1. Predicted Probabilities

To validate the prediction models, for each patient an individual predicted probability was calculated by integrating the following linear part prediction formulas developed by Sweerts et al., in 1/(1 + exp^−linear part^) × 100% [2]: Surgical site infection: −7.272 + (0.031 × age − 0.002 × BMI + 0.757 × smoking status + 0.891 × immunological disorder + 0.904 × diabetes mellitus + 2.345 × liver disease + 0.619 × NSAID’s);Postoperative bleeding: −7.172 + (0.033 × age + 0.012 × BMI − 0.023 × smoking status + 0.729 × heart disease + 0.787 × vitamin K antagonist use);Delirium: −14.307 + (0.127 × age + 0.348 × heart disease + 0.898 × disease of central nervous system);Nerve damage: −2.250 + (−0.051 × age − 0.254 × gender + 0.572 × smoking status − 0.009 × dysplasia).

#### 2.6.2. Predicted Probabilities

The overall model performance was expressed by the distance between the predicted and actual outcome [27]. To quantify model performance, the Brier statistic was determined. For the Brier statistic, squared differences between the actual outcome and predictions were calculated. The Brier statistic can range from 0 for a perfect model to 0.25 for a non-informative model with 50% incidence of the outcome [28]. The ability of the model to discriminate between patients with and without the outcome was assessed using the area under the curve (AUC). The AUC can range from 50% (no discriminative capacity) to 100% (perfect discriminative capacity). The discriminative capacity was considered moderate when AUC was > 0.70 and good when AUC was > 0.80 [29]. Calibration of the model is the agreement between predicted probabilities (probability of an event calculated with the model) and observed frequencies of outcome (accuracy) and was assessed by visually inspecting the calibration plot [27]. Furthermore, we computed Hosmer and Lemeshow (H-L) goodness-of-fit as a quantitative measure of calibration. A high H-L statistic is related to a low *p*-value, and indicates a poor fit [30]. All statistical analyses were performed using R 3.5.3 and its extension packages vim, mice, rms, pROC, and generalhoslem [31].

## 3. Results

A total of 2641 medical records of patients who received THA or TKA were included. Of these, 1407 patients received a primary THA and 1207 patients received a primary TKA. 

Patient characteristics of the study cohort are shown in Table 1. The mean age was 68 years and 62% of the patients were female. 

The automated versus manual search resulted in an agreement of κ = 0.94 for the extraction of the structured data. A κ = 0.55 was found for the extraction of the unstructured data. 

### 3.1. Model Development

The number of missing values per predictor variable is shown in Table 1. For the majority of the predictors, there were no missing data. Missing data were found for BMI (1.2%) and smoking status (1.3%). Analysis showed that the data were missing at random. After multiple imputation, all data of all patients were available for analysis.

### 3.2. Model Performance

The ROC curves representing the discriminative performance of the prediction models are shown in Figure 1. The corresponding AUCs are reported in Table 2. 

The mean predicted probability for surgical site infection was 1.3% (range 0.2–33.5%). For postoperative bleeding, delirium, and nerve damage, the mean predicted probabilities of, respectively, 1.5% (range 0.2–8.6%), 0.8% (range 0.01–11.8%), and 0.3% (range 0.05–4.1%) were found, see also Table 3. The predictive performances of the models are shown in the calibration plots in Appendix A. The model for surgical site infection showed an overestimation exceeding 2% risk for surgical site infection. The model for postoperative bleeding showed an underestimation of the actual risk, and the model for nerve damage showed poor calibration overall. For delirium, moderate calibration was found, resulting in an underestimation of the actual probability between 2 and 6% and exceeding 8%. The H-L statistic showed *p*-Values < 0.001 for all models, which indicates a poor fit.

## 4. Discussion

The aim of this study was to externally validate four previously developed models for preoperative counselling by predicting the risk for surgical site infection, postoperative bleeding, delirium, and nerve damage in patients after THA and TKA. External validation showed only good performance for delirium. Calibration of the model for delirium was acceptable, and the discriminative capacity was good with an AUC 95%-lower limit confidence interval of 0.82. For all other models, calibration was poor, resulting in under- or overestimation, and discriminative capacity was poor to moderate. The results for the models for surgical site infection, postoperative bleeding, and nerve damage showed diminished accuracy when used on another population than originally developed for, and do not provide reliable estimations of the predicted probabilities to be used in preoperative counselling and shared decision-making. Overall, external validation showed a loss of discriminative performance; the original AUC for the model for surgical site infection was 72% instead of 55% for the model in external validation. The same corresponds for postoperative bleeding; 73% vs. 61%, delirium; 86% vs. 84%, and nerve damage; 77% vs. 57% [2].

Only three studies with three procedure-specific prediction models (including the models which we externally validated in this study) to predict surgical site infection are available in the literature for comparison with our results [2,14,15]. To our knowledge, these procedure-specific models have not been externally validated previously [18]. For a prediction of postoperative bleeding, delirium, and nerve damage, no other procedure-specific models for preoperative counselling in THA or TKA have been found.

Previous research has shown that the external validation of prediction models often results in poorer performance [32], and that external validation using different populations is negatively influenced by differences in centers (geographical validation) [33]. The same may have been true for this external validation. The differences in patients between the cohorts of the academic and non-academic hospitals may have led to the poor results for the prediction of surgical site infection, postoperative bleeding, and nerve damage.

The model of delirium showed moderate discriminative capacity with three included predictors. Only this model was found to be appropriate for clinical use and we consider this result important for preoperative counselling and shared decision-making. The model consists of three predictor variables only (age, heart disease, and disease of central nervous system), and as such, may be easy to use since the predictors are considered to be known in usual preoperative care. Being able to predict postoperative delirium based on three predictor variables is arguably useful in clinical practice to take early precautions to prevent or treat delirium.

All in all, a plethora of prediction models for the prediction of surgical complications are available—universal models, procedure-specific models, models with many variables, and models with a smaller amount of variables—but a common problem is that numerous models after external validation seem to have difficulties regarding discrimination and calibration and thereby clinical applicability to (another) specific population [34]. In this case, it can be recommended to adjust or recalibrate a model for local circumstances by the use of information of the primary model with information of the validation study [34,35].

### Strengths and Limitations

We used a reproducible, automated method to extract the data from the electronic health records, and we cross-checked the automated search with a manual search. The cross-check revealed very good agreement between the automated and manual extraction of the structured data (κ = 0.94). A moderate agreement (κ = 0.55) was found for the data extraction of the unstructured data. Checking the results showed that the automated search with the clinical data collection tool extracted the unstructured data more accurately than the researcher because the former was searching in more sources (e.g., medical notes and evaluations reported by all medical specialists within a predefined timeframe) while the manual extraction was limited to orthopedic and preoperative anesthesia data. Additionally, we selected a representative cohort of patients, as shown by the comparable frequencies of postoperative complications within our (academic hospital) reference cohort and the current (non-academic hospital) cohort.

This study has a number of limitations. We collected retrospective data using a clinical data collection tool. This tool is considered a promising tool for retrieving real-world data from electronic health records because relevant outcome data can be identified [36]. However, data extracted from this system only represent (real-world) data that are entered in the electronic patient records. If these latter data are incomplete, erroneous, or missing, they will also (negatively) influence the retrieved data. For example, we found several typographical errors in the physicians’ notes and different physicians used different terminology for the same diseases; this will arguably have negatively influenced the data extractions. On the other hand, the clinical data collection tool assisted in mitigating these issues by proposing a combination of keywords, commonly known synonyms, word variations, abbreviations, common typographical errors, and by checking data for accuracy using a built-in validation tool. Furthermore, we tried to prevent discrepancies and errors by employing a sensitive search first and specifying the query by adding information while continuously confirming that no data loss took place. Although we tried to prevent errors with multiple activities, we cannot rule out having missed registered comorbidities in text fields, which may have resulted in an underestimation of the frequencies of comorbidities. Another limitation is our low number of events regarding the surgical outcomes, particularly for nerve damage. We cannot ensure this to be of influence on the results of this external validation, especially regarding model calibration. Different to other studies, the model for delirium did not include smoking status and gender as predictor candidates for the prediction of delirium; this is in contrast with other studies [37,38]. In the developmental study, predictor candidates were selected based on evidence from the literature, clinical reasoning, and eyeballing potential higher frequencies in the data [2]. The chosen method of inclusion has not led to the inclusion of these potential predictors based on the developmental cohort.

## 5. Conclusions

This study externally validated four prediction models that are aimed to improve preoperative counselling and shared decision-making at the orthopedics department. Only the model for delirium showed good discriminative capacity and calibration to be appropriate for clinical use. The results for the models for surgical site infection, postoperative bleeding, and nerve damage suggest that these models do not provide sufficient predictive accuracy to be applied in clinical settings. Taking the effective ways to prevent and/or threaten delirium into account, we expect the model for the prediction of delirium to be valuable for preoperative counselling, shared decision-making, and early delirium precautionary interventions. This expectation is strengthened by the fact that this model included only age, the preoperative presence of a heart disease, and the presence of a disease of the central nervous system as predictor variables, thus encompassing the proven important ease of use by keeping the data entry to a minimum [19]. Studies assessing the utility of these models are needed to explore if these prediction models can improve counselling efforts and have practical benefits.

## Figures and Tables

**Figure 1 jpm-13-00277-f001:**
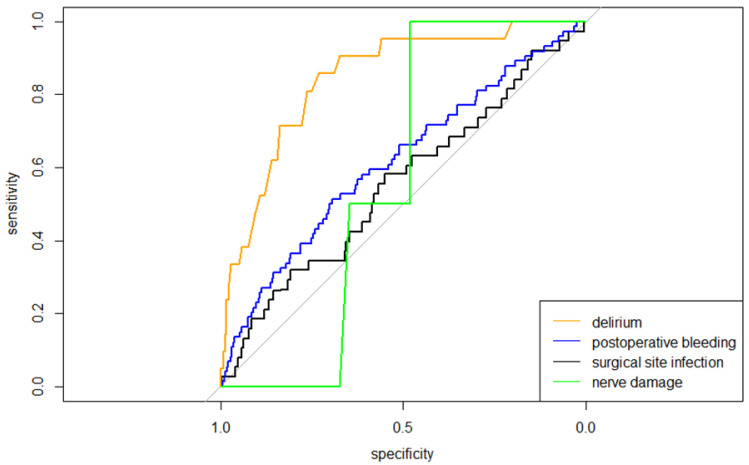
ROC curves of all four models predicting surgical complications, indicating the discriminative performance of all models concerning the probability of a surgical complication.

**Table 1 jpm-13-00277-t001:** Patient characteristics of study cohort.

Patient Characteristics	Missing Values	Total Cohort (*n* = 2614)	Patients after THA (*n* = 1407)	Patients after TKA (*n* = 1207)
Age (years, mean ± SD)	0%	68.1 ± 10	68.8 ± 10.5	67.4 ± 9.4
Gender: female (n, %)	0%	1628 (62.3)	892 (63.4)	736 (61)
BMI (mean ± SD)	1.20%	28.8 ± 5	27.7 ± 4.8	30.1 ± 5.1
Smoking: yes (n, %)	1.30%	522 (20)	304 (21.9)	218 (18.2)
Surgical complications (n, %)				
-surgical site infection	0%	38 (1.5)	23 (1.6)	15 (1.2)
-postoperative bleeding	0%	74 (2.8)	40 (2.8)	34 (2.8)
-delirium	0%	21 (0.8)	8 (0.6)	13 (1.1)
-nerve damage	0%	-	2 (0.1)	-
Comorbidities (n, %)				
-immunological disorder	0%	316 (12.1)	152 (10.8)	164 (13.6)
-rheumatoid arthritis	0%	205 (7.8)	101 (7.2)	104 (8.6)
-diabetes mellitus	0%	348 (13.3)	159 (11.3)	189 (15.7)
-liver disease	0%	41 (1.6)	24 (1.7)	17 (1.4)
-heart disease	0%	622 (23.8)	342 (24.3)	280 (23.2)
-disease of central nervous system	0%	145 (5.5)	76 (5.4)	69 (5.7)
-hip dysplasia	0%	39 (1.5)	36 (2.6)	3 (0.2)
Medication use				
-vitamin K antagonist	0%	151 (5.8)	87 (6.2)	64 (5.3)
-NSAID	0%	296 (11.3)	189 (13.4)	107 (8.9)

Abbreviations: BMI, body mass index; NSAID, non-steroid anti-inflammatory drugs.

**Table 2 jpm-13-00277-t002:** Discriminative (AUC) and predictive (H-L) performance per model.

Discriminative and Predictive Performance	Area under the Curve (AUC) (95%CI)	H-L Statistic (*p*-Value)
Surgical site infection	0.55 (0.52–0.58)	<0.001
Postoperative bleeding	0.61 (0.59–0.64)	<0.001
Delirium	0.84 (0.82–0.87)	<0.001
Nerve damage	0.57 (0.53–0.61)	<0.001

Abbreviations: AUC, area under the curve: the ability to discriminate between those with and without the outcome. The AUC can range from 0.50 (no discriminative capacity) to 1.00 (perfect discriminative capacity). H-L statistic, Hosmer and Lemeshow: quantitative measure of calibration. High H-L statistic is related to a low *p*-Value and indicates a poor fit.

**Table 3 jpm-13-00277-t003:** Mean predicted risk, and Brier statistic per model.

Overall Performance	Mean Predicted Risk % (SD)	Brier Statistic
Surgical site infection	0.013 (0.022) Range 0.002–0.335	0.015
Postoperative bleeding	0.015 (0.012) Range 0.002–0.086	0.028
Delirium	0.008 (0.011) Range <0.001–0.118	0.008
Nerve damage	0.003 (0.003) Range 0.001–0.041	0.001

Brier statistic: to quantify model performance. Squared differences between actual outcome and predictions are calculated. The score can range from 0 for a perfect model to 0.25 for a non-informative model with 50% incidence of the outcome [27].

## Data Availability

Data are available upon reasonable request.

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
