# Peer review of "External Validation of Prediction Models for Surgical Complications in People Considering Total Hip or Knee Arthroplasty Was Successful for Delirium but Not for Surgical Site Infection, Postoperative Bleeding, and Nerve Damage: A Retrospective Cohort Study"

_jpm, 2023, doi:10.3390/jpm13020277_

Round 1

Reviewer 1 Report

The manuscript by Sweerts et al. reports external validation results for four previously developed models for prediction of surgical complications in individuals considering primary total hip or total knee arthroplasty. Among the four models, only one (delirium) demonstrates good external validity whereas the others perform poorly compared to the development study. Regardless, all presented external validation results are of great interest considering the future development of these models and potential clinical use of the risk prediction model for delirium. Overall, the manuscript is well written and easy to follow. All sections contain relevant information considering the study aims and the selected statistical methods are sound to reach them. My recommendation is accept after consideration of the following minor remarks for the manuscript:

1) There seems to be an error of the caption of Table 2 as it does not reflect the content of the table and is identical to Table 1.

2) The number of one of the outcomes of interest, nerve damage, is very low. Consider mentioning it as a limitation in the external validation of the corresponding model.

3) In the discussion, it is stated that "External validation showed only moderate performance for delirium". However, in my opinion, considering the low number of events, also the calibration plot looks good as the grouped observations fall very close to the line of identity. Perhaps the tone of of these validation findings could be a bit more favourable? I believe that if there would be more of these outcomes available, the nonparametric calibration curve would look substantially better as well. One could consider mentioning the low number of these events as a limitation as well, especially in terms of model calibration.

Reviewer 2 Report

The present manuscript submitted by Sweets Lieke et al. addresses the external validation of four internally validated prediction models for surgical complications after THA and TKA. Around 2,614 patients cohort was selected for this study who underwent primary THA or TKA in secondary care between 2017 and 2020. Individual predicted probabilities of the risk for surgical complication (i.e., surgical site infection, postoperative bleeding, delirium, and nerve damage) per outcome were calculated for each model. The authors concluded that Out of four prediction models, Only the model for delirium showed good discriminative capacity and calibration to be appropriate for clinical use. The models for surgical site infection, postoperative bleeding, and nerve damage suggest do not provide sufficient predictive accuracy to be applied in clinical settings. Therefore, the Authors recommend to clinicians, use this simple and straightforward delirium model during preoperative counseling, shared decision-making, and early delirium precautionary interventions.

In my opinion,  the manuscript was well written, technically sounds perfect, and was successful to convince the aim of the study. The authors also tried to point out the limitation of this study while handling the clinical data. Overall, the information and sample size for the prediction model was sufficient and valuable, and well presented in this manuscript.

Minor comments:

Authors must explain the importance of predicted probabilities of the risk of surgical complication i.e surgical site infection, postoperative bleeding, delirium, and nerve damage after THA and TKA. Why do these risks need to be addressed before surgery which is missing in the introduction?

Is there any difference or change in BMI before and after THA and TKA in Patients?
